# Towards an Efficient Data Fragmentation, Allocation, and Clustering Approach in a Distributed Environment

**Hassan Abdalla [1],\* and Abdel Monim Artoli [2]**

[1]   College of Technological Innovation, Zayed University, P.O. Box 144534, Abu Dhabi 11543, UAE
[2]   College of Computer and Information Sciences, King Saud University, P.O. Box 51178,
     Riyadh 11543, Saudi Arabia; aartoli@ksu.edu.sa
\*   Correspondence: Hassan.Abdalla@zu.ac.ae

**Abstract:** Data fragmentation and allocation has for long proven to be an efficient technique for improving the performance of distributed database systems' (DDBSs). A crucial feature of any successful DDBS design revolves around placing an intrinsic emphasis on minimizing transmission costs (TC). This work; therefore, focuses on improving distribution performance based on transmission cost minimization. To do so, data fragmentation and allocation techniques are utilized in this work along with investigating several data replication scenarios. Moreover, site clustering is leveraged with the aim of producing a minimum possible number of highly balanced clusters. By doing so, TC is proved to be immensely reduced, as depicted in performance evaluation. DDBS performance is measured using TC objective function. An inclusive evaluation has been made in a simulated environment, and the compared results have demonstrated the superiority and efficacy of the proposed approach on reducing TC.

**Keywords:** vertical fragmentation; clustering; data allocation; data replication; site clustering; DDBS

## 1. Introduction

Despite the fact that numerous distributed database systems' (DDBS) design methods have been broadly presented in the recent years, there still exist many challenges that need to be addressed for improving the quality of DDBS design, particularly those concerned with transmission costs (TC). An increasing interest is still directed towards finding an efficient DDBS design that guarantees a maintainable high performance in DDBSs. One of the crucial components of performance evaluation in a distributed environment is the amount of data being transmitted over network sites while queries are being addressed. Several previous works addressed this issue (known in literature as transmission cost); however, very few have come up with a clear and decisive measure for DDBS performance evaluation.

As a matter of fact, the challenging task of finding an efficient DDBS design is steadily driving research in this field. On one extreme, this intriguing trend in research has led to the production of a variety of different methods/techniques with the basic aim of promoting distribution productivity. On the other extreme, the proliferation of these techniques has led to more confusion as it comes to select which one is more effective than the other, at least from the distribution point of view. Nevertheless, there has been also constant consensus on the principles and concepts, which underlie these methods and techniques, by which each technique is to be evaluated [1]. Researchers, from DDBS design and distributed computing domains, have been presenting several approaches to tackle the challenges of the design of distributed systems. Some of these approaches are extended and enhanced to integrate

different combination of tools to improve DDBS performance [2–8]. The findings of these approaches are reinforced by testing them on either synthetized data, verified in a simulated environment, or on real datasets, or even on both. In this work, we propose a data fragmentation and allocation approach with the sole emphasis on reducing transmission cost (TC) to the minimum. The suggested approach is oriented on developing data fragmentation, utilizing a site clustering algorithm, and suggesting new data allocation scenarios. Moreover, an extensive evaluation has been made in comparison with the work of [8], as it is the closest work for the present work of this paper. The evaluation results show the undeniable enhancement on DDBS performance based on TC minimization.

The contributions of this paper include: (1) Developing a data fragmentation algorithm based on an agglomerative hierarchical clustering, with the aim of reducing the number of iterations needed to perform hierarchical clustering and finding solution space; (2) developing a site clustering technique that seeks to produce a minimum number of high-balanced clusters; (3) introducing advanced scenarios for data allocation and examining them in detail to find the best fitting one; and (4) finally, examining the effectiveness of the data replication impact on DDBS performance through profoundly-done internal and external evaluations.

The remainder of the paper is structured as follows. Section 2 provides a thorough investigation of the recent closely-related works. Section 3 presents the proposed methodology. A practical experiment is vividly illustrated in Section 4. Section 5 draws in-depth performance evaluation and vigorous discussion on the obtained results. Finally, conclusions and future work directions are summarized in Section 6.

## 2. Related Work

In DDBS design, it is consensually agreed-upon that the more precise the data partitioning and allocation techniques, the better the performance and the lower the response time are likely to be obtained [8]. In [8], a comprehensive taxonomy was given. This taxonomy was fine-grained and comprehensively analyzed in both static and dynamic environments. The main issues addressed in this taxonomy includes data fragmentation, data allocation, and replication. These issues have been examined to classify and analyze a huge number of previously-made DDBSs works. The observation of earlier works' drawbacks was the drive aim of this taxonomy to produce more productive methods to improve DDBS performance. It was found that TC minimization (including communication costs) has been the key objective, for which most of old and recent works have been striving to achieve by maximizing data locality and minimizing remote data access. Nevertheless, it was noted in taxonomy that most of these works failed to provide a clear definition for TC as a performance metric, which is considered as a huge shortcoming.

An improved system to fragment data at the initial stage of DDBS design and allocate fragmented data at runtime over the cloud environment was presented in [4]. A Cloud Based Distributed Database System (CB-DDBS) architecture over a cloud environment was developed. CB-DDBS adopted a replication scenario so that DDBSMs are allowed to work in parallel to meet the client's orders. Even though the proposed algorithm of CB-DDBS was hugely driven by the Performance Optimality Enhancement Algorithm (POEA) [9], authors had never indicated this inspiration. Moreover, selecting a cluster leader was not practical enough to work in a real-world environment as most DDBSs have the same specification for all of its members (nodes), specifically in the Peer-2-Peer network. On the other hand, the data replication problem (DRP) was deeply addressed by [6] and formulated as an integer linear problem, with an assumption of having overlapping horizontally-divided data. In fact, the replication problem was looked at as an optimization problem to gain the intended aim of having fragments' copies and sites kept at a minimum. On the other extreme, [10] drew a method based on the particle swarm optimization (PSO) algorithm to shorten TC. The core of this study was to solve a data allocation problem (DAP) by utilizing the PSO algorithm. Fragments allocation over sites had been done with the PSO algorithm, and its performance was evaluated on 20 different tested problems.

On the same line, in [2], an enhanced version of [3] was developed. The work sought to incorporate a site clustering algorithm, for network sites, and a mathematical replication model for cost-effective data allocation [3]. A significant enhancement was observed in terms of overall DDBSs performance through decreasing TC among network sites. The constraints of clusters and sites were also taken into account to strengthen the proposed efficiency. In-depth experiments were carried out to solely prove the effectiveness of this technique, with respect to minimizing TC and promoting DDBS performance. As a matter of fact, an evidential reduction in TC and clear enhancement in DDBSs performance had been demonstrated. Additionally, this work was profoundly evaluated against [3], with respect to the objective function of [2]. Results positively proved that [2] far outperformed [3], in terms of decreasing TC and significantly increasing the overall DDBS productivity.

On the other hand, [11] used a Genetic Algorithm (GA) with Mahalanobis distance along with the K-means clustering algorithm as an influential combination to propose a two-phase clustering algorithm for distributed datasets. In first phase, GA is utilized in parallel on fragments, which were assigned to different sites. Mahalanobis distance was used as fitness value in GA. To draw a better representation of initial data, covariance between data points were considered in Mahalanobis distance. In second phase, to find final results, K-means with K-means++ initialization was applied on intermediate output. To conduct experiments and measure performance, multiple real-life and synthetic datasets were used to implement the technique in the framework of Hadoop.

On the same page, in [5], an enhanced vertical fragmentation approach was presented using the K-means Rough (KR) clustering technique. Several experiments were conducted and results using KR showed that: (1) the smaller the number of clusters K, the larger the total time and the satisfactory error average cost and memory cost K-means algorithm were obtained; (2) the larger the number of clusters of k, the more optimized were these three criteria in comparison with the normal k-Means algorithm. Lastly, in [12,13], the authors proposed an approach of a greedy nature to fragment and allocate fragments in DDBS. While they used an aggregated similarity to cluster similar queries and find fragments, they were planning to use a greedy algorithm to assign resulted fragments into relative sites. However, they neither demonstrated the approach mechanism nor conducted experiments to verify the approach effectiveness.

In the proposed work of this paper, a comprehensive approach is released for the purpose of finding the best fitting technique for DDBS design. The intended technique is meant to further minimize TC while giving an obvious definition for TC. It is worth mentioning that the criteria involved in taxonomy [7] are also considered in this paper. TC is being lessened by increasing data locality and decreasing remote data access, while communication overhead is significantly reduced by adopting replication scenarios.

## 3. Methodology

The proposed architecture and heuristics of this approach are clearly depicted in Figure 1. A five-phase process was proposed as follows: In the first phase, for data fragmentation, queries (Qs) under consideration were set to produce (N) disjointed fragments, using hamming distance based on a hierarchical clustering process. The proposed refinement process, in second phase, was meant to draw non-overlapping schemes from overlapping schemes produced from the first phase. The fragmentation evaluator (FE) of [14] was employed with the aim of having the non-overlapping partitioning schema. This schema is the survival schema as a result of applying FE on all disjoint schemes. In the third phase, network sites were set to be grouped using the proposed clustering method. In the fourth phase, a data allocation process was activated to work on the survival schema, so that schema was set to be assigned to network clusters/sites in accordance with the data allocation model. The data allocation process was bound to be activated competitively over (CN) clusters of sites, and (M) sites at each cluster. These cluster were the results of applying the proposed clustering algorithm on network sites, as will be explained in the site clustering section. Finally, in the fifth phase (as per the proposed

data allocation model), all scenarios were thoroughly examined under different circumstances so that the highest TC-reducing scenario was going to be selected and incorporated into the DDBS design.

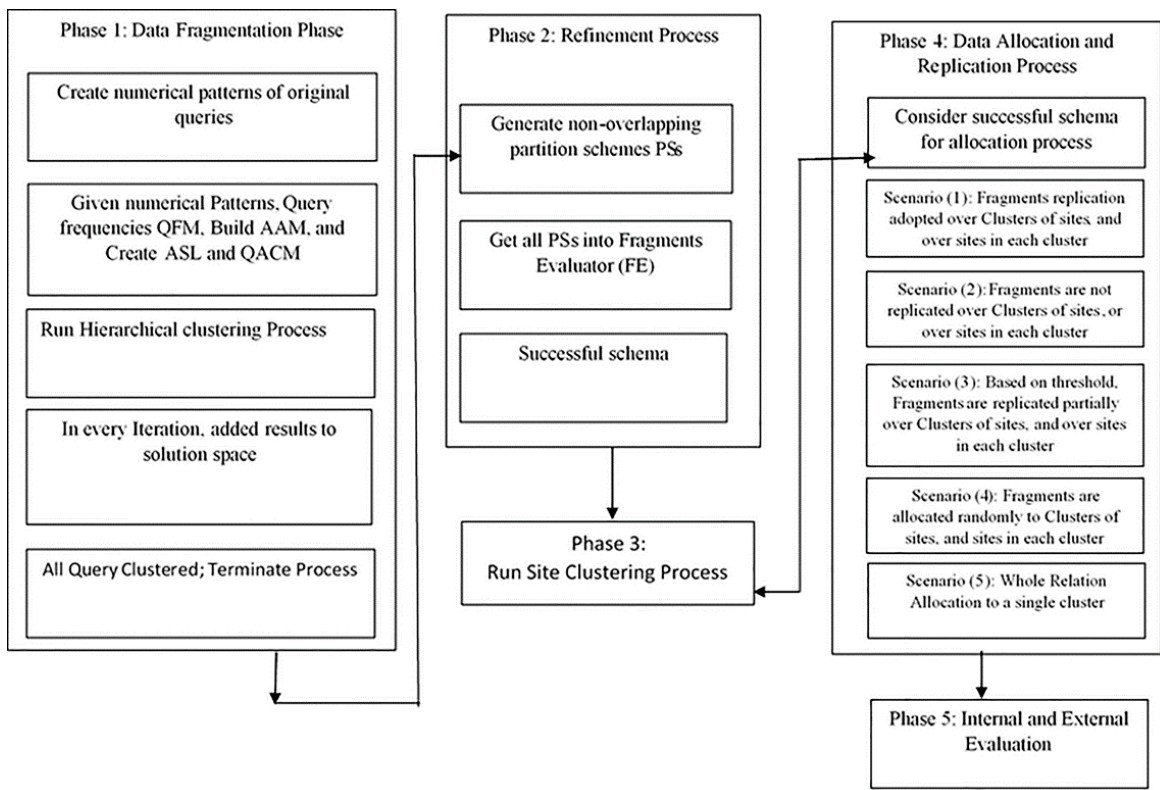

**Figure 1.** Overview of vertical fragmentation and allocation heuristics.

*3.1. Fragmentation Cost Model*

3.1.1. Objective Function

Where TFQ is the total frequency of each (k) query over all (M) sites, and XF is the binary constant, which indicates 1 if the fragment (F) is placed in the site (S); and 0, otherwise. COM is communication costs across clusters of sites (CCM) or within sites in each cluster (CSM). On the other hand, selectivity (Q) is a rate of query k over fragment F, and is equal to ($|Al_{ij}|/NA_{ik}$), where $|Al_{ij}|$ is the attributes number in $F_i$, which are locally reached by $Q_k$, and $NA_{ik}$ is the entire number of attributes in fragment $F_i$ distantly accessed, in regard to $F_j$, by $Q_k$. Finally, size (F) is the size of the fragments under consideration in bytes.

$$\text{Func(TC)} = \text{Minimize}\left( \sum_{k=1}^{q} \sum_{j=1}^{m} \text{TFQ}_{kf} * \text{XF}_{kj} * \text{COM}_{s_i s_j} * \{\text{Selectivity }(Q_k) * \text{size}(F_k)\} \right), \, f = 1,..,n \qquad (1)$$

3.1.2. Cost Functions

Firstly, the model used a query set, supposedly taken from workload, to build an attribute query matrix (AQM). Each $aqm_{ij}$ refers to attribute $A_i$ as it is contained by query $Q_k$. A query frequency matrix (QFM) was assumed to be given by the database administrator (DBA), so each $qfm_{ij}$ provided how many times each query was being released from its relevant site $S_j$. Using these requirements, the model of vertical fragmentation was set to be working as per the following functions:

$$\text{TFQ}_{kx} = \sum_{k=1}^{q} \sum_{j=1}^{m} \text{QFM}_{kj}, \qquad x = 1 \qquad (2)$$

$$AUM_{ki} = \sum_{j=1}^{m} \sum_{k=1}^{q} TFQ_{kx} * AQM_{kj}, \qquad x = 1 \tag{3}$$

$$ASL_k = \sum_{i=1}^{n} \sum_{k=1}^{q} AUM_{ki} \tag{4}$$

$$QACM_{ik} = \sum_{i=1}^{n} \sum_{k=1}^{q} ASL_k \tag{5}$$

$$Sim\,(Q_{k1}, Q_{k2}) = \sum_{k1=1}^{q} \sum_{k2=1}^{q} (1 - dif(P((Q_{k1}), \text{Numerical Pattern}(Q_{k2})) \tag{6}$$

$$QDM_{k1k2} = \sum_{k1=1}^{q} \sum_{k2=1}^{q} Sim\,(Q_{k1}, Q_{k2}) \tag{7}$$

Where (Sim) stands for similarity. These functions; however, are further explained in the self-explanatory steps in the results section.

### 3.1.3. Fragmentation Evaluator (FE)

To evaluate schemes, this work employs the FE, which was drawn in [14]. It has two metrics: relevant remote access and irrelevant local access. The relevant remote access calculates the net access costs of remote attributes stored at other sites, different from the site from which a query was issued. On the other hand, the irrelevant local access associates with attributes that were observed by local processing. Equation (8) computes the first term of FE, the square-error of the entire partition scheme containing a certain number of "nf" fragments:

$$E_{nf}^2 = \sum_{i=1}^{nf} \sum_{q=1}^{Q} \left[ TFQ * |Al_{ik}| \left(1 - \frac{|Al_{ik}|}{NA_i}\right) \right] \tag{8}$$

where NA is the attributes number of the targeted relation. On the other hand, Equation (9) seeks to provide the second term of FE, as it computes the ratio of remote attributes being accessed:

$$E_{ad}^2 = \sum_{i=1}^{nf} \sum_{q=1}^{Q} \left[ \sum_{j=1}^{m} TFQ_{qi}^i * |Ad_{ik}| * \left(\frac{|Ad_{ik}|}{NA_{iqk}}\right) \right] \tag{9}$$

where |AD| is the attributes number in Fi, which is remotely reached with regard to Fj, by Qk. Hence, FE is given by its two metrics as follows:

$$FE = E_{nf}^2 + E_{Ad}^2 \tag{10}$$

As a rule of thumb in FE evaluation, the lower the FE value is, the better the DDBS performance is and vice versa.

### 3.2. Clustering Methodology

### 3.2.1. Query Merging

The numerical patterns of queries are used to merge queries carefully in clusters, using which, fragments are bound to be produced. To calculate the differences between patterns as a similarity function, hamming distance [15] was employed. These patterns are then gathered using the least difference value (LDV) concept, Equations (6) and (7).

### 3.2.2. Sites Grouping

In this work, sites were clustered by applying the concept of least difference value (LDV). In its turn, LDV concept was used to group queries in [7], and, in fact, proved to be highly effective in improving DDBS performance. LDV seeks to cluster points based on their least distance/difference. Moreover, the LDV-based clustering was shown to outperform the threshold-based algorithm, which was applied to cluster sites in [7]. Threshold-based clustering has sometimes been observed to either shrink cluster numbers to an unpractical range or increase this number to an undesirable extent [2], which leads, in both cases, to an unavoidable negative impact on DDBS rendering, as described in the discussion section.

Problem Formulation

Given a set of sites S $\{S_1, S_2, \ldots, S_m\}$, the clustering solution Cs is defined as a set of clusters $C_1$, $\ldots, C_t$ which draws the partitioning (clustering) of sites S. So, we can say that we had $C \subseteq S, \forall C_i \in C$, $C_i \subseteq C; C_i \sqcup C_{i+1} \sqcup C_{i+2} \sqcup \ldots \sqcup C_t = C; I = 1..t; \forall C_i$ and $C_j; C_i \sqcap C_j = \ominus$, where $\ominus$ is empty { }, as we seek to produce disjointed clusters.

Clustering Algorithm

Initialization: Given a set of sites M, the communication costs matrix between sites, initial clusters were initiated using the LDV concept.

Loop: For any new site, the following was done:

1. Calculate communication costs between the new site and each cluster using the average communication costs. Average costs will be used as a decisive membership for each site with respect to clusters under consideration.
2. The cluster of the lowest average cost is bound to be the candidate container for the site at hand.
3. If more than one candidate container is recorded, the container of the lowest distance to the targeted site is the primary and sole container.
4. Repeat steps (1–3) until all sites are clustered successfully.

It is important to point out that, as opposed to a threshold-based algorithms [7,16], executing a clustering algorithm, as per steps (1–4), ensured that no outliers could be produced due to the fact that all sites were clustered. Consequently, this achievement of the proposed algorithm was observed to be a great contribution, as no site/cluster could be lost. On the other hand, to perform a fair comparison between the proposed work of this paper and [7], Table 1 provides the same communication costs matrix, which is exclusively used for the first experiment in [7]. This experiment was done separately for illustration purposes.

**Table 1.** Communication costs between sites [7].

| Site# | Site 1 | Site 2 | Site 3 | Site 4 | Site 5 | Site 6 |
|-------|--------|--------|--------|--------|--------|--------|
| Site 1 | 0 | 10 | 8 | 2 | 4 | 6 |
| Site 2 | | 0 | 7 | 3 | 5 | 4 |
| Site 3 | | | 0 | 3 | 2 | 5 |
| Site 4 | | | | 0 | 11 | 5 |
| Site 5 | | | | | 0 | 5 |
| Site 6 | | | | | | 0 |

### 3.3. Allocation and Replication Model

### 3.3.1. Requirements

Given attribute set A = $\{A_1, A_2, ..., A_n\}$ required by query set Q = $\{Q_1, Q_2, ..., Q_k\}$, these queries were bound to be grouped into (Q) query clusters $\{Cq_1, Cq_2, \ldots., CQ_{cn}\}$. Then query clusters were

going to be allocated to M sites S = {$S_1$, $S_2$, . . . ., $S_m$}, which were already grouped into $C_m$ clusters of sites, Cs = {$Cs_1$, $Cs_2$, . . . ., $CSc_m$} in a fully-connected network. Let $F$ = {$F_1$, $F_2$, ..., $F_n$} be the set of disjointed partitions/fragments, which were obtained from fragmentation process. The allocation model struggled to observe the optimal distribution of each partition/fragment (F) over clusters Cs, and consequently on the sites of clusters individually.

### 3.3.2. Allocation Scenarios and Phases

#### Scenario 1 (Full Replication); Phase 1

Each fragment was going to be assigned to all clusters of sites. On the other hand, inside each cluster, a fragment wass given for one single site.

#### Scenario 2 (Non-Replication); Phase 1

Each fragment was assigned to the cluster of the highest total access cost (TACC). This cost was used as a controller to assign fragments over clusters. However, TACC is the maximum cost required to reach fragments' attributes separately.

#### Phase 2 for Both Scenarios 1 and 2

For each site, the net cost, which is a site's total access cost ($TACS_{ij}$) to access $F_i$, was computed and used as s decisive measure by which $F_i$'s allocation decision is taken. As per data cost functions (Equation (12)), the TACS matrix is constructed using Attribute Access Matrix (AAM) and Communication Cost Matrix (CMS). Using the TACS matrix, a fragment was set to be allocated to the site of the highest access value, as shown in the results section.

#### Mixed Scenario 3 (Hybrid Replication, Full and Partial)

Each fragment is set to be assigned to all clusters as the full replication scenario is being adopted. Fragments are given for more than one single site in each cluster as per the precisely calculated threshold. This threshold; however, could be simply calculated by taking the average of maximum and minimum transmission costs in each cluster. Then, each site that succeeds to surpass the threshold is a candidate for holding a copy of the fragment in question. This scenario proves to be highly efficient, mainly due the fact that updated queries grow slowly and slightly, reflecting the real-world DDBS behavior. However, if update queries grow quickly and significantly (which is rare in the real distribution environment), this scenario would be proven to be unsuitable.

### 3.3.3. Cost Functions

While Equation (11) computed the attribute access matrix of sites (AAMS), AAMS was used to yield the total access cost matrix for all sites (TACS) with the help of Equation (11). In Equations (12) and (13), the final allocation of fragments over the cluster of sites was decided when second and third scenarios of allocation were being addressed. Finally, for data replication, this work adopted the same model presented in [7]. Table 2 describes the site constraints, represented in virtual capacity (in Megabyte), of the lower and upper allowed attribute limits for each site. LA and UA stand for lower/Upper limit of attributes was allowed for each site to have.

$$AAMS = \sum_{k=1}^{q} \sum_{j=1}^{m} \sum_{i=1}^{n} AQM_{ik} * QFM_{ji} \tag{11}$$

$$TACS = \sum_{j1=1}^{m} \sum_{i=1}^{n} \sum_{j=1}^{m} AAMS_{ij1} * CMS_{ji} \tag{12}$$

$$TACC = \sum_{j1=1}^{m} \sum_{i=1}^{n} \sum_{j=1}^{m} AAMS_{ij1} * CCM_{ji} \tag{13}$$

**Table 2.** Site constraints.

| Site | $S_1$ | $S_2$ | $S_3$ | $S_4$ | $S_5$ | $S_6$ |
|------|-------|-------|-------|-------|-------|-------|
| Capacity (MB) | 15,500 | 12,200 | 9500 | 10,000 | 11,100 | 15,000 |
| LA | 1 | 1 | 1 | 1 | 1 | 1 |
| UA | 10 | 12 | 12 | 6 | 12 | 12 |

## 4. Practical Experiment

This experiment was conducted using a machine with a Intel core i3 duo processor, with a speed of 2.80 GHz, 4 GB RAM, and a Windows 7, 32-bit operating system. C-sharp language was used to analyze and interpret data given as initial input represented in the queries, query frequency matrix (QFM) and communication costs matrix. To highlight the contribution of this work in terms of TC reduction, we made an external evaluation with [7]. We created an environment that ensured a fair comparison between our work of this paper and [7]. In other words, the same simulation environment in which [7] was been implemented, was also adopted in this paper. To illustrate the mechanism of our approach, one experiment was performed. The virtual network was assumed to be fully-connected network of sites, as shown in Figure 2.

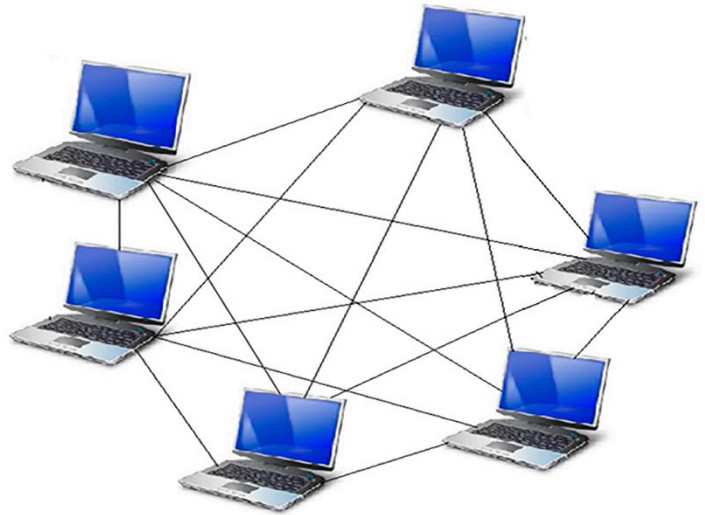

**Figure 2.** Network site of fully-connected architecture.

It worth mentioning here that, in the first experiment, the proposed employee (Emp) dataset was created in accordance with the description provided in Table 3. For the first experiment, the dataset had six attributes and 300 rows. For the sake of simplifying computation and completeness, the schema and attributes were re-drawn in this work. The Emp-no, Emp-name, Job-id, Salary, Location and Dept-id, were referred to as A1, A2, A5, A6, A5, and A6, respectively.

**Table 3.** Emp database description.

| Attributes | Symbol | Type | Length (Bytes) |
|------------|--------|------|----------------|
| Emp-no | A1 | Nominal | 4 |
| Emp-name | A2 | Categorical | 30 |
| Job-id | A3 | Categorical | 4 |
| Salary | A4 | Numerical | 3 |
| Location | A5 | Categorical | 5 |
| Dept-id | A6 | Nominal | 4 |

For this experiment, it was assumed that the queries under consideration were eight queries running against the dataset, say 50% to 85% of all queries.

Q1: Select A1, A2, A5, A6    from Emp where A1 in (1234, 261, 1239) and A3 = "Mang222";

Q2: Select A3, A5             from Em where A5 in ('site 1', 'site 3', 'site 6');

Q3: Select A2, A4, A5         from Em;

Q4: Select A1, A3, A6         from Em where A6 = 'dept2';

Q5: Select A1, A2, A5         from Emp where A2 = "Jane" and A5 in ('site 2','site 5');

Q6: Select A3, A4, A6         from Emp where A4 > 4500;

Q7: Select A2, A6            from Emp where A1 > 1234;

Q8: Select A1, A3, A5, A6   from Emp;

## 4.1. Implementation

To activate the clustering process, attribute incidences were taken as the initial input. The attribute query matrix (AQM) was formed so that each $AQM_{kj}$, Table 4, showed in which query ($Q_k$) the attribute ($A_j$) was contained.

**Table 4.** Attribute incidence matrix (AIM).

| Query/Attribute | A1 | A2 | A3 | A4 | A5 | A6 |
|:---:|:---:|:---:|:---:|:---:|:---:|:---:|
| Q1 | 1 | 1 | 0 | 0 | 1 | 1 |
| Q2 | 0 | 0 | 1 | 0 | 1 | 0 |
| Q3 | 0 | 1 | 0 | 1 | 1 | 0 |
| Q4 | 1 | 0 | 1 | 0 | 0 | 1 |
| Q5 | 1 | 1 | 0 | 0 | 1 | 0 |
| Q6 | 0 | 0 | 1 | 1 | 0 | 1 |
| Q7 | 0 | 1 | 0 | 0 | 0 | 1 |
| Q8 | 1 | 0 | 1 | 0 | 1 | 1 |

Additionally, QFM (Table 5) was needed to perform clustering, presumably provided by DB administrator. Each value ($QFM_{kj}$) indicated the number of times each query $Q_k$ was issued from its original site $S_j$.

**Table 5.** Query frequency matrix (QFM).

| Query/Site | S1 | S2 | S3 | S4 | S5 | S6 | TQF |
|:---:|:---:|:---:|:---:|:---:|:---:|:---:|:---:|
| Q1 | 2 | 0 | 0 | 0 | 1 | 3 | 6 |
| Q2 | 2 | 2 | 0 | 0 | 3 | 0 | 7 |
| Q3 | 0 | 0 | 3 | 3 | 0 | 0 | 6 |
| Q4 | 0 | 0 | 0 | 1 | 0 | 3 | 4 |
| Q5 | 0 | 2 | 0 | 0 | 0 | 2 | 4 |
| Q6 | 0 | 1 | 1 | 3 | 0 | 0 | 5 |
| Q7 | 2 | 0 | 0 | 1 | 0 | 0 | 3 |
| Q8 | 0 | 1 | 1 | 0 | 2 | 1 | 5 |

Where TQF stands for total of query frequency. Then, using the fragmentation cost model, drawn above, the query difference matrix (QDM) was produced as shown (Table 6). Each $QDM_{ij}$ represented the difference value between numerical patterns of $Q_i$ and $Q_j$. A number of initial clustering was undertaken in the algorithm in every phase of its cycle.

**Table 6.** Query difference matrix (QDM).

| Query | Q1 | Q2 | Q3 | Q4 | Q5 | Q6 | Q7 | Q8 |
|-------|----|----|----|----|----|----|----|----|
| Q1 | 0 | 4 | 3 | 3 | 1 | 5 | 2 | 2 |
| Q2 | 4 | 0 | 3 | 3 | 3 | 3 | 4 | 2 |
| Q3 | 3 | 3 | 0 | 6 | 2 | 4 | 3 | 5 |
| Q4 | 3 | 3 | 6 | 0 | 4 | 2 | 3 | 1 |
| Q5 | 1 | 3 | 2 | 4 | 0 | 6 | 3 | 3 |
| Q6 | 5 | 3 | 4 | 2 | 6 | 0 | 3 | 3 |
| Q7 | 2 | 4 | 3 | 3 | 3 | 3 | 0 | 4 |
| Q8 | 2 | 2 | 5 | 1 | 3 | 3 | 4 | 0 |

*4.2. Hierarchical Clustering Process*

QDM, in its turn, was used as basic values for the clustering process. Tables 7 and 8 exhibit the final results of applying the hierarchical clustering on QDM.

**Table 7.** QDM (final results).

| Query | Q1573 | Q4826 |
|-------|-------|-------|
| Q1573 | 0 | 2 |
| Q4826 | 2 | 0 |

**Table 8.** Final distribution of queries on clusters.

| Cluster/Query | Q1 | Q2 | Q3 | Q4 | Q5 | Q6 | Q7 | Q8 |
|---------------|----|----|----|----|----|----|----|----|
| CQ1357 | 1 | | 1 | | 1 | | 1 | |
| CQ2468 | | 1 | | 1 | | 1 | | 1 |

It was obvious that the clustering process needs four, |eight/two|, loops to have all solutions added to the solution space, (Table 9).

**Table 9.** Solution space.

| Solution # | Cluster # | Queries Contained |
|------------|-----------|-------------------|
| Solution 1 | Cq1 | Q1573 Q2468 |
| Solution 2 | Cq2 | Q12357 Q468 |
| Solution 3 | Cq3 | Q13578 Q246 |
| Solution 4 | Cq4 | Q13567 Q248 |

*4.3. Refinement Process*

The overlapping partitioning schemes (PS) produced in the fragmentation phase were further refined. A process was used to properly examine each PS attribute, attribute by attribute, to decide their belonging within the PS, in which it would be allocated, Table 10. As the decisive factor for allocating attributes over query clusters, the attribute allocation decision (AAD) was drawn in Equation (14).

$$AAD = \begin{cases} C_k, & A_i \text{ highly accessed by } C_k \text{ for each } C_k, \ k = 1, \ldots, \text{ NC cluster} \\ \text{Otherwise} & A_i \text{ replicated interchangeably over partitions} \end{cases} \tag{14}$$

As mentioned earlier, in this phase, the refinement process looked to to secure non-overlapping schemes from those overlapping, which were already drawn in the solution space. For each partitioning schema (PS), each shared attribute $A_i$, such as $A_1$, $A_4$, $A_5$, and $A_6$ in solution $QC_1$, was examined among all partitions of the same solution. Normally, attributes will be assigned to the partition with the highest access to it on the basis of its call in the query set. However, if $A_i$ is equally required by

more than one partition, it shall be assigned interchangeably to all partitions in the same partitioning schemes (PSs). That accounts for yielding four schemes from $QC_1$. In doing such placement iteratively, all schemes combinations were set to be generated, as given in Table 10. On the other hand, for each non-shared attribute $A_i$, such as $A_1$ in $QC_1$, this attribute would be kept to its container, as produced from the fragmentation process. Moreover, during the refinement process, it seemed that we may have had a schema duplication, thus only one copy was set to be kept. The rationale behind performing this process in such a pattern is to maintain all possible schemes that could lead to optimal survival schema.

**Table 10.** Refinement process results.

| PS Number | Over-Lapping PS | Non-Overlapping PS |
|---|---|---|
| 1 | | (A1,A2,A4,A5) (A3,A6) |
| 2 | QC1 | (A2,A5) (A1,A3,A4,A6) |
| 3 | (A1,A2,A4,A5,A6) (A1,A3,A4,A5,A6) | (A1,A2,A5) (A3,A4,A6) |
| 4 | | (A2,A4,A5) (A1,A3,A6) |
| 5 | QC3 | (A1,A2,A4,A5,A6) (A3) |
| 6 | (A1,A2,A5,A6,A4,A3) (A3,A5,A1,A6,A4) | (A1,A2,A5,A6) (A3,A4) |
| 7 | QC4 (A1,A3,A5,A6,A4,A2) (A3,A5,A6,A1) | (A2,A4,A5,A6) (A1,A3) |

### 4.4. Fragmentation Evaluator (FE)

To decide which schema is the optimal, all non-overlapping PS under consideration were fed into FE. Figure 3 depicts, briefly, the results of the FE process.

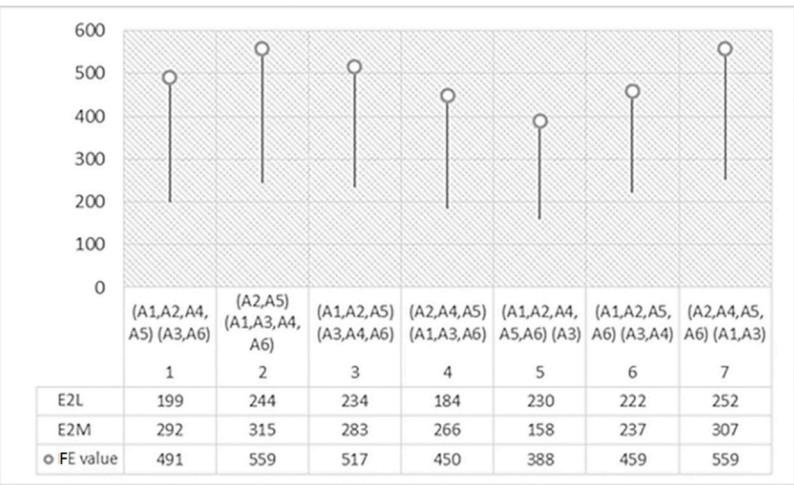

**Figure 3.** Partitioning schemes.

From Figure 3, the successful schema was PS5, and its attribute allocation to query clusters is given in Table 11.

**Table 11.** Attribute allocation over query clusters.

| Cluster/Attribute | A1 | A2 | A3 | A4 | A5 | A6 |
|---|---|---|---|---|---|---|
| CQ1 | | 1 | 1 | | 1 | 1 | 1 |
| CQ2 | | | | 1 | | | |

The next SQL statements were set to in order to produce the final fragments:

DataF1 = C1: Select A1,A2,A4,A5,A6        from Employee; size (DataF1) = 13,800 bytes.

DataF2 = C2: Select A3        from Employee; size (DataF2) = 1200 bytes.

### 4.5. Allocation Process

The first step was to produce the attribute access matrix of sites (AAMS) using QFM and AIM along with Equation (11). Every $AAMS_{ij}$ gave the net access cost of each site $S_j$, to reach Attribute $A_i$, Table 12.

**Table 12.** Attribute access matrix of sites (AAMS).

| Site/Attribute | A1 | A2 | A3 | A4 | A5 | A6 |
|:---:|:---:|:---:|:---:|:---:|:---:|:---:|
| S1 | 2 | 4 | 2 | 0 | 4 | 4 |
| S2 | 3 | 2 | 3 | 1 | 5 | 2 |
| S3 | 1 | 3 | 2 | 4 | 4 | 2 |
| S4 | 1 | 4 | 4 | 6 | 3 | 5 |
| S5 | 3 | 1 | 5 | 0 | 5 | 2 |
| S6 | 9 | 5 | 4 | 0 | 6 | 7 |

Using Equation (12), AAMS was multiplied with the communication cost matrix between sites to form the total access cost matrix (TACS), Table 13.

**Table 13.** Total access cost matrix (TACS).

| Site/Attribute | A1 | A2 | A3 | A4 | A5 | A6 |
|:---:|:---:|:---:|:---:|:---:|:---:|:---:|
| S1 | 106 | 86 | 98 | 54 | 144 | 96 |
| S2 | 81 | 98 | 87 | 46 | 126 | 107 |
| S3 | 91 | 85 | 79 | 25 | 116 | 100 |
| S4 | 94 | 59 | 94 | 15 | 120 | 77 |
| S5 | 81 | 101 | 91 | 79 | 112 | 120 |
| S6 | 49 | 72 | 79 | 54 | 104 | 77 |

### 4.6. Data Allocation: Scenario (1), (Fragments are Replicated over Site Clusters)

For [7], Table 14 briefs the fragment allocation over all clusters.

**Table 14.** Decision allocation matrix, [7].

| Sites' Cluster/Fragment | | F1 | | | | | | F2 | |
|:---:|:---:|:---:|:---:|:---:|:---:|:---:|:---:|:---:|:---:|
| | Sites/Attributes | A1 | A2 | A5 | A4 | A6 | Total Cost (F1) | A3 | Total Cost (F2) |
| | S1 | 106 | 86 | 144 | 54 | 96 | 486 | 98 | 98 |
| CS1 | S2 | 81 | 98 | 126 | 46 | 107 | 458 | 87 | 87 |
| | S3 | 91 | 85 | 116 | 25 | 100 | 417 | 79 | 79 |
| CS2 | S4 | 94 | 59 | 120 | 15 | 77 | 365 | 94 | 94 |
| | S5 | 81 | 101 | 112 | 79 | 120 | 493 | 91 | 91 |
| CS3 | S6 | 49 | 72 | 104 | 54 | 77 | 356 | 79 | 79 |

For the proposed work of this paper, Table 15 briefs the fragment allocation over all clusters.

**Table 15.** Decision allocation matrix of this work.

| Sites' Cluster/Fragment | | F1 | | | | | | F2 | |
|---|---|---|---|---|---|---|---|---|---|
| | Sites/Attributes | A1 | A2 | A5 | A4 | A6 | Total Cost (F1) | A3 | Total Cost (F2) |
| CS1 | S6 | 49 | 72 | 104 | 54 | 77 | 356 | 79 | 79 |
| | S2 | 81 | 98 | 126 | 46 | 107 | 458 | 87 | 87 |
| CS2 | S4 | 94 | 59 | 120 | 15 | 77 | 365 | 94 | 94 |
| | S1 | 106 | 86 | 144 | 54 | 96 | 486 | 98 | 98 |
| CS3 | S3 | 91 | 85 | 116 | 25 | 100 | 417 | 79 | 79 |
| | S5 | 81 | 101 | 112 | 79 | 120 | 493 | 91 | 91 |

4.6.1. The Second Allocation Scenario (No Fragment Replication)

For [7], Table 16, holds total of Communication Access for Clusters (TACC), and 17 brief the fragment allocation over all clusters.

**Table 16.** TACC decision allocation matrix, [7].

| Cluster # | A1 | A5 | A2 | A4 | A6 | Total Cost of F1 | A3 | Total Cost of F2 |
|---|---|---|---|---|---|---|---|---|
| C1 | 857.5 | 920 | 1332 | 546 | 1074.5 | 4730 | 1042.5 | 1042.5 |
| C2 | 1218 | 1301.5 | 8171 | 770.5 | 1445.5 | 12,906.5 | 1319 | 1319 |
| C3 | 2265 | 2146 | 3090 | 1135 | 2500 | 11,136 | 2245 | 2245 |

After that, the competitive process was activated in order to have fragments assigned to sites in each cluster. As drawn in the data allocation model, each fragment would be allocated to the site of the highest access cost, Table 17.

**Table 17.** Fragment allocation over sites (Scenario 2), [7].

| Sites' Cluster/Fragment | | F1 | | | | | | F2 | |
|---|---|---|---|---|---|---|---|---|---|
| | Sites/Attributes | A1 | A2 | A5 | A4 | A6 | Total Cost (F1) | A3 | Total Cost (F2) |
| CS1 | S1 | 106 | 86 | 144 | 54 | 96 | 486 | 98 | 98 |
| | S2 | 81 | 98 | 126 | 46 | 107 | 458 | 87 | 87 |
| | S3 | 91 | 85 | 116 | 25 | 100 | 417 | 79 | 79 |
| CS2 | S4 | 94 | 59 | 120 | 15 | 77 | 365 | 94 | 94 |
| | S5 | 81 | 101 | 112 | 79 | 120 | 493 | 91 | 91 |
| CS3 | S6 | 49 | 72 | 104 | 54 | 77 | 356 | 79 | 79 |

For the proposed work of this paper, Tables 18 and 19 brief the fragment allocation over all clusters.

**Table 18.** TACC fragments allocation over site clusters of our proposed work.

| Cluster # | A1 | A2 | A4 | A5 | A6 | Total Cost of F1 | A3 | Total Cost of F2 |
|---|---|---|---|---|---|---|---|---|
| C1 | 1460 | 1365 | 727 | 1932 | 1619 | 7103 | 1426 | 1426 |
| C2 | 906 | 1068 | 612 | 1374 | 1212 | 5172 | 1008 | 1008 |
| C3 | 1250 | 1285 | 707 | 1942 | 1439 | 6623 | 1406 | 1406 |

**Table 19.** Decision allocation matrix of this work.

| Sites' Cluster/Fragment | | F1 | | | | | | F2 | |
|---|---|---|---|---|---|---|---|---|---|
| | Sites/Attributes | A1 | A2 | A5 | A4 | A6 | Total Cost (F1) | A3 | Total Cost (F2) |
| CS1 | S6 | 49 | 72 | 104 | 54 | 77 | 356 | 79 | 79 |
| | S2 | 81 | 98 | 126 | 46 | 107 | 458 | 87 | 87 |
| CS2 | S4 | 94 | 59 | 120 | 15 | 77 | 365 | 94 | 94 |
| | S1 | 106 | 86 | 144 | 54 | 96 | 486 | 98 | 98 |
| CS3 | S3 | 91 | 85 | 116 | 25 | 100 | 417 | 79 | 79 |
| | S5 | 81 | 101 | 112 | 79 | 120 | 493 | 91 | 91 |

### 4.6.2. The Hybrid (Mixed) Allocation Scenario (Full Fragment Replication over Clusters, Partially over Sites in Each Cluster)

This technique is markedly used as an integrated set-up for both works. In Phase (1), data were allocated to all clusters using the "full replication principal". Each fragment had to be allocated to each cluster, as shown in Table 20.

**Table 20.** Phase 1, final allocation over cluster of sites in both works.

| Cluster # | A1 | A5 | A2 | A4 | A6 | Total Cost of F1 | A3 | Total Cost of F2 |
|---|---|---|---|---|---|---|---|---|
| C1 | 857.5 | 920 | 1332 | 546 | 1074.5 | 4730 | 1042.5 | 1042.5 |
| C2 | 1218 | 1301.5 | 8171 | 770.5 | 1445.5 | 12,906.5 | 1319 | 1319 |
| C3 | 2265 | 2146 | 3090 | 1135 | 2500 | 11,136 | 2245 | 2245 |

In Phase (2), in each cluster, every site that exceeded (surpass) the threshold value was going to contain the qualified fragments, as shown in Tables 21 and 22.

**Table 21.** Fragment allocation over sites (Scenario 3), [7].

| Sites' Cluster/Fragment | | F1 | | | | | | F2 | |
|---|---|---|---|---|---|---|---|---|---|
| | Sites/Attributes | A1 | A2 | A5 | A4 | A6 | Total Cost (F1) | A3 | Total Cost (F2) |
| CS1 | S1 | 106 | 86 | 144 | 54 | 96 | 486-surpass | 98 | 98-surpass |
| | S2 | 81 | 98 | 126 | 46 | 107 | 458-surpass | 87 | 87-surpass |
| | S3 | 91 | 85 | 116 | 25 | 100 | 417 | 79 | 79 |
| CS2 | S4 | 94 | 59 | 120 | 15 | 77 | 365 | 94 | 94 |
| | S5 | 81 | 101 | 112 | 79 | 120 | 493 | 91 | 91 |
| CS3 | S6 | 49 | 72 | 104 | 54 | 77 | 356 | 79 | 79 |

**Table 22.** Fragment allocation over sites (Scenario 3) of this work.

| Sites' Cluster/Fragment | | F1 | | | | | | F2 | |
|---|---|---|---|---|---|---|---|---|---|
| | Sites/Attributes | A1 | A2 | A5 | A4 | A6 | Total Cost (F1) | A3 | Total Cost (F2) |
| CS1 | S6 | 49 | 72 | 104 | 54 | 77 | 356 | 79 | 79 |
| | S2 | 81 | 98 | 126 | 46 | 107 | 458 | 87 | 87 |
| CS2 | S4 | 94 | 59 | 120 | 15 | 77 | 365 | 94 | 94 |
| | S1 | 106 | 86 | 144 | 54 | 96 | 486 | 98 | 98 |
| CS3 | S3 | 91 | 85 | 116 | 25 | 100 | 417 | 79 | 79 |
| | S5 | 81 | 101 | 112 | 79 | 120 | 493 | 91 | 91 |

In some cases, the process of assigning fragments over sites (in each cluster) was found to be very similar in both works, as the full replication scenario and hybrid scenario were addressed as shown in Tables 15 and 22. In other words, both scenarios had almost the same impact on DDBS performance.

## 5. Results and Discussion

We evaluated the performance of this work by conducting several experiments, among which just eleven are presented in this section for demonstration purposes. Relation cardinality, number of sites, number of queries, and rate of query types were all varied in each experiment. In doing so, both our proposed work and that of [7] were set to be examined under different circumstances. As mentioned earlier, this work had the aim of increasing data locality to the greatest possible extent so that TC was going to be maximally reduced. In other words, data fragments were placed in the cluster/site where it was highly and frequently required. As consequences, transmission costs (TC), including communication costs, and response time were substantially mitigated. To validate and verify these claims, internal and external evaluations were made. For the sake of ensuring a fair comparison, we tried to create the same environment in which [7] had been tested. Thus, for first part of this evaluation, the same five problems addressed in [7] were also considered in this work, namely queries 8, 16, 24, 30, and 40, respectively. While the first problem was separately done in both works, it was exclusively restricted for retrieval queries (read-type). The second and third problems considered a mixture of retrieval and update queries, but with retrieval queries having a larger portion. Finally, the last two problems also were a mixture, but with update queries taking a larger share. The evaluation process was made in terms of many design-related performance factors. Among these factors was (1) the TC reduction rate, which is of paramount importance to be investigated; and (2) DDBS performance, which is calculated as (1—averaged TC), where averaged TC is the averaged costs incurred as the query set of the certain experiment is being processed. That is, performance had a inversely proportional relationship with TC in this work. For each problem, meeting the minimization of (TC) along with the objective function was considered.

For the first problem, Figures 4–7 show the experiments that were carried out and which reflect the clear contribution of work of this paper. While Figures 4 and 5 display TC rate for both works as per site clustering of [8]; Figures 6 and 7 show results of TC rate when applying the new proposed site clustering of this paper, on both [7] and our work of this paper. According to the results obtained, our work proved to be highly efficient with respect to TC minimization. Every query, among those under consideration, was tested on employee dataset in accordance to five data allocation scenarios: (1) hybrid replication-based Scenario (HAS); (2) full replication Scenario (FAS); (3) no replication over clusters of sites Scenario (NAS); (4) random allocation Scenario; and (5) random allocation for the whole dataset. These experiments were done using C#, to determine which fragmentation and allocation scenario gave better results for DDBS performance. It is worth indicating that we refer to reference [7] as [7] and [Adel et al, 2017] interchangeably, and we refer to the proposed work of this paper as "present" (in figures) to facilitate comparison and make it clearly understood.

It was clearly evident form Figures 4 and 5 that scenario (1) outperformed its peers, particularly when communication cost between clusters was considered, Figure 7. While scenario (2), for the first problem, came in second place with a slight difference followed by scenario (3). Scenario (5), on the other hand, was recorded to be the worst for DDBS performance. Needless to say, it was possibly true that scenario (1) was the best for the first three problems, since all data were available in all clusters and were of read type queries that had the larger space of queries under consideration. These facts were not surprising since it agreed with results observed in [7]. However, in the proposed work of this paper, as per results on all drawn figures, TC reduction was clearly observed to be greatly lessened when compared to [7]. In the sense that it can be confidently deduced that the enhancement of the present work proved to be highly valuable and effective.

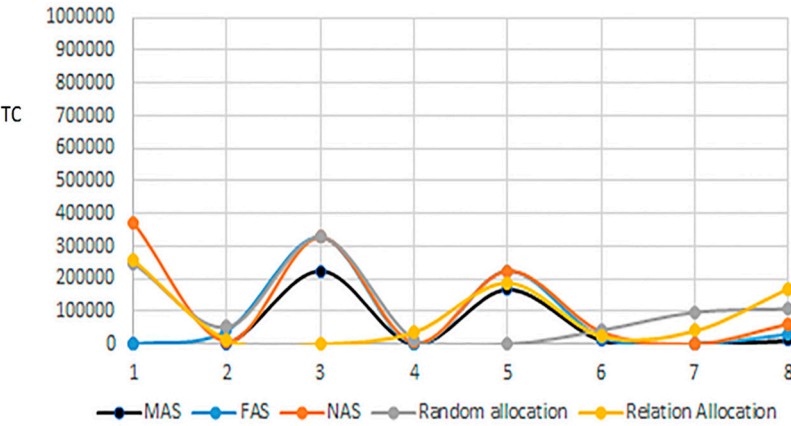

**Figure 4.** Problem 1, [7], transmission costs with site clustering of [7].

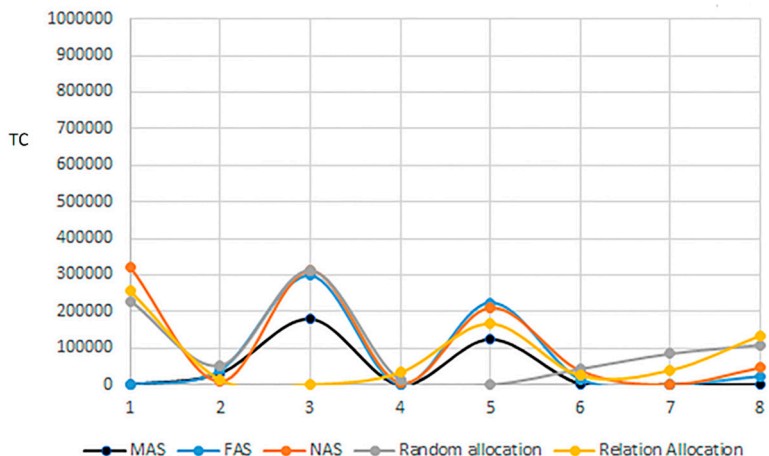

**Figure 5.** Problem 1, present work of this paper, transmission costs with site clustering of [7].

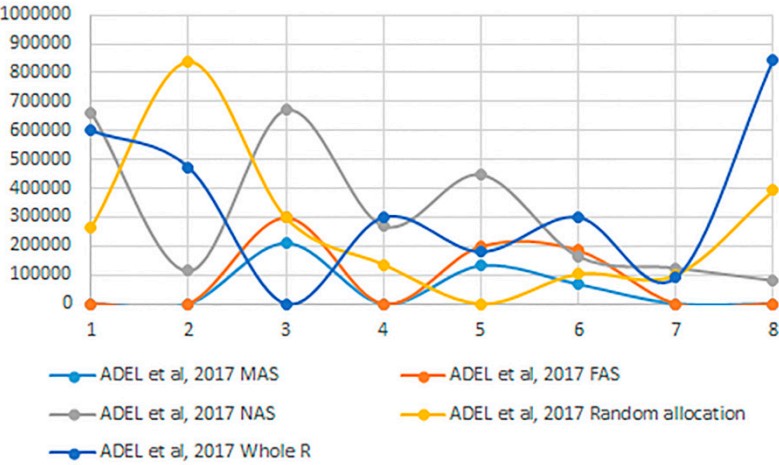

**Figure 6.** Problem 1, [7] transmission costs with site clustering of our proposed work.

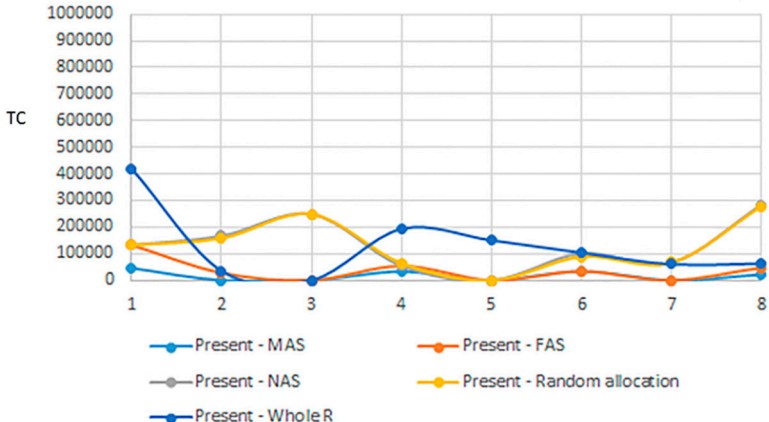

**Figure 7.** Problem 1, transmission costs with site clustering of this work.

In [7], communication costs between clusters were taken as the average of all points of intended clusters. This justifies why [7] recorded great results in this work. In other words, comparing [7] in Figure 4 with [7] in Figure 5, it was evidential that the LDV-based clustering process for network sites showed to be highly effective in terms of TC reduction. On the other hand, in most cases, our work of this paper showed to perform better than [7] in all scenarios of communication, as shown in Figures 6 and 7.

From the results shown in Figures 4–7, it can be concluded that data replication had a huge impact on communication cost minimization, mainly when retrieval queries established the largest portion of the considered queries. To emphasize this claim, four more experiments (P1, P2, P3, and P4) were performed with 16, 24, 30, and 40 queries, respectively. The obtained results presented in Figures 8 and 9 confirmed that scenarios (1) and (2) were the best scenarios, whereby retrieval queries represented the largest portion of considered queries, as they were in P1, P2, and P3. However, scenario (3), followed by scenario (4), were considered to be, by far, the best options when updated queries constituted the largest percentage of the queries under consideration, as they were in P4 and P5.

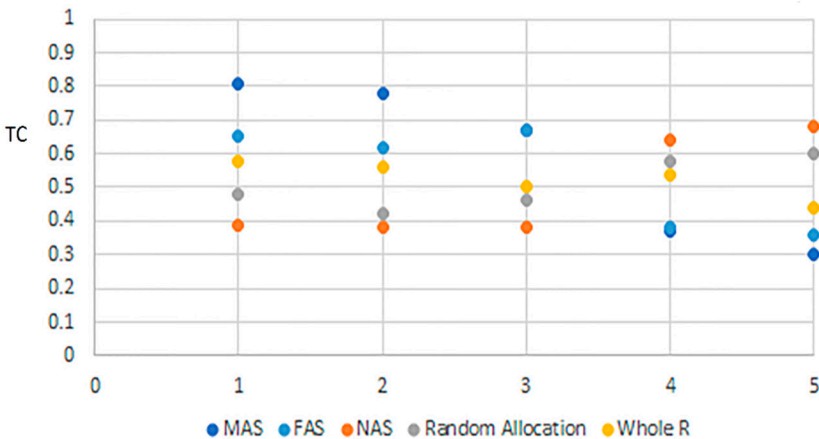

**Figure 8.** Problem 1, TC minimization in percentage over all scenarios in [7].

Our proposed site clustering technique played a key role in balancing clusters and thus in increasing data compactness, locality, and availability in each cluster. These factors, in fact, contributed highly in reducing TC and heightening DDBS performance, as revealed in Figures 10 and 11. All problems described previously were evaluated in the same pattern in which problem (1) was examined. It was clear from the results illustrated in Figures 10 and 11 that our proposed method performed better than [7], for all problems tackled and all considered scenarios regarding TC minimization.

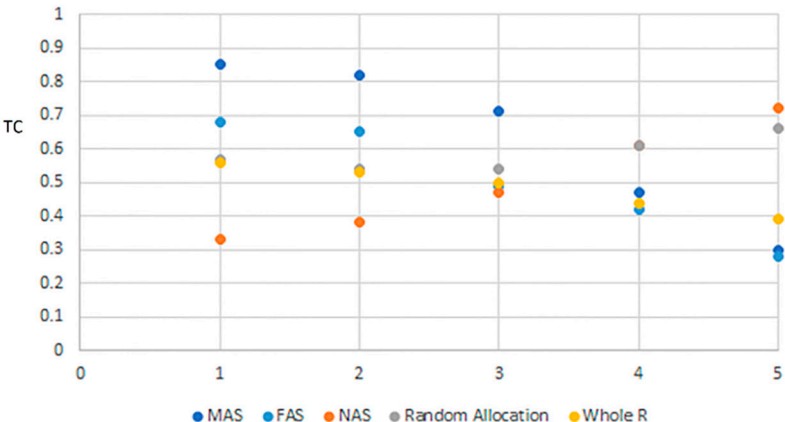

**Figure 9.** Problem 1, TC minimization in percentage for all scenarios for our proposed work.

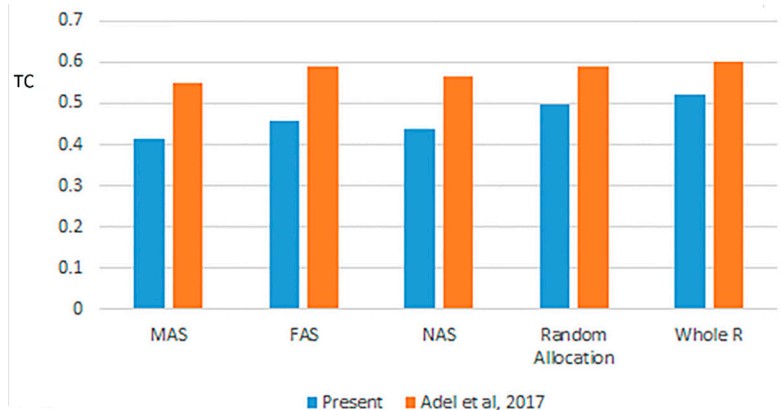

**Figure 10.** TC for the five problems, with all scenarios of the works under comparison.

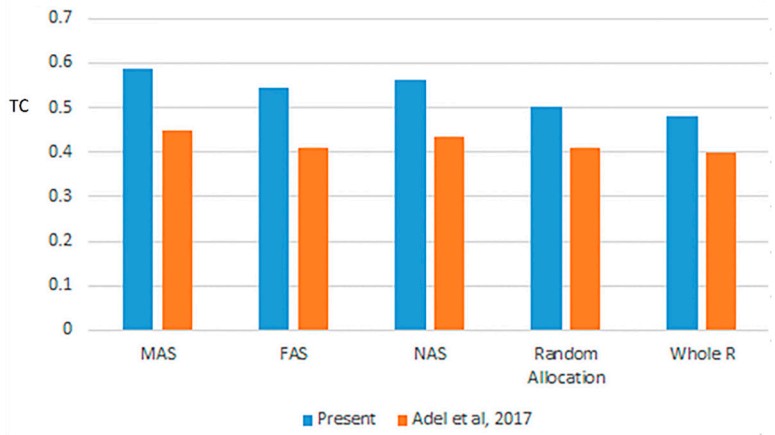

**Figure 11.** Performance for the five problems, with all scenarios of the works under comparison.

*Approach Complexity of Computation*

Similarly, like [7] and [17], the complexity of time was bounded by O (NF*QN*M), at the best case, and O (N2), at the worst case. Where NF, QN, M and N stand for Number of Fragments, Number of Query clusters, number of sites and number of queries respectively.

## 6. Conclusions and Future Work Directions

In DDBSs, the issue of performance sustainability is an energetic issue that needs extra investigation. As a matter of fact, several factors have been logged in literature that could have a great impact on DDBS performance. Among these factors are data fragmentation, data allocation,

replication, and site clustering, which are crucial factors in determining DDBS performance. However, the objective of maintaining a high performance could effectively be achieved through clustering the frequently-accessed attributes together, and placing them in closely-related sites/clusters so that they significantly match the requirements of their corresponding queries/applications.

Therefore, in this work, a heuristic approach was elegantly introduced. The proposed approach considerably reduced transmission costs of distributed queries, as verified in the results and discussion section. The presented fragmentation procedure was accomplished based on a cost-effective model that was set to be used in the context of a relational database, at initial and later stages of DDBS design. On the other hand, the site clustering algorithm was made in such way that ensures the production of highly-balanced clusters with proven significance and efficiency. Moreover, this work suggests several advanced allocation scenarios that take data replication into consideration, including full replication, partial replication, and non-replication.

During this work, several experiments were thoroughly conducted to reinforce the superiority of the proposed approach, specifically the site clustering algorithm compared to previous similar algorithms [7].

Five data allocation scenarios were considered in this work, three of them were replication-based allocations. The five scenarios were: mixed replication-based data allocation scenario (MAS), full-replication-based data allocation scenario (FAS), and non-replication data allocation scenario (NAS). To show how considerable of a negative or positive impact data replication could have, a workable method for these scenarios was performed using the proposed objective function. Several experiments under different circumstances were carried out to select the best TC-reducing design in DDBS environment. It was observed that data replication had an obvious negative impact on DDBS productivity where update queries were growing. In comparison with [7], based on the obtained results, the proposed work proved to behave better for about 76% of all experiments that were carried out in this work. However, in some rare cases [7] was shown to perform better.

Our future work is going to be in the same direction with the aim of finding a better-response time optimization technique in the cloud environment.

**Author Contributions:** Conceptualization, H.A.; Methodology, A.M.A. and H.A.; Data Analysis, H.A.; Manuscript Writing, H.A.; Validation, A.M.A.

**Funding:** This work has been supported by the Research Centre at the College of Computer and Information Sciences, King Saud University.

**Conflicts of Interest:** The author declares no conflict of interest.

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
