# Peer review of "Towards an Efficient Data Fragmentation, Allocation, and Clustering Approach in a Distributed Environment"

_information, doi:10.3390/info10030112_

Round 1

Reviewer 1 Report

The problem of fragmenting and allocating Data is as old as DDBSs. In early seventies, R. G. Casey, W. W. Chu, K. P. Eswaran, E. Holler, R. Peebles, V. K. M. Whitney among others proposed methods and solutions to optimize data allocation and minimize data exchange among nodes of a computer network. These works were based on the computer technology of the time, mainly constituted of mainframes and slow wired networks. With the advent of mini and personal computers the interest in such problems tended to diminish, but the introduction of cloud systems revitalized them in view of the new storage and transmission capabilities of distributed systems and they became a front research area again. 

This paper is a follow-up of a previous one, co-authored by the author, in which improvements are claimed with respect to the data fragmentation, clustering, allocation and replication algorithms. However, the conclusion of a large experimental work is rather discouraging and it reaches facts that are known  since longtime such as the approximate 20%/80% updates/reads limit for data replication and some performance improvements w.r.t. the results of the previous work. However I could not check the content of the previous work since it is not publicly accessible.

Moreover the writing of the paper suffers of some sloppiness:

- line 145 -  the sentence "Finally, in fifth phase, as..." seems to be incomplete

- line 189 -  "Query.Frequency ..." no dot is needed

- line 191 -   "... query released ..." lacks the verb

- line 258 -  "... as there is no site/cluster..." delete "there is"

- line 290 -  the sentence is badly structured

- Figures 10 and 11 -  the vertical axis is labelled TC in both figures, so what do you mean by DDB Performance in Figure 11? I could not find a clear definition of DDB Performance before; is it the average query response time??

- the bibliographical references are a mess. Some entries are listed with first name first, some with first name initial, some with last name first. Reference 13 has no title. In the text most references are listed by number, but some by [Author, year]. In particular, I supposed that the most referenced item (i. e. the previous work) is both referenced as [8] and [Adel et al, 2017], but [Adel et al, 2017] could also correspond to [2] and multiple references correspond to [Amer, 2018].

In conclusion I suggest the author either to produce a conference paper only highlighting the improvements to the previous work or to re-write the paper for a journal in a self consistent form (without continuously referring to the previous paper), possibly putting in an Appendix all the cumbersome intermediate passages and tables, but evidentiating the starting hypotheses and the simulation results.

Author Response

Dear respected Editor-in-chief and Unknown Reviewers,

I would like to thank you very much for giving me the opportunity to revise my manuscript based on your reviewers’ invaluable recommendations and suggestions. I am highly grateful for the time and energy they have put to review this work. I carefully went through their suggestions and made the necessary amendments to the manuscript considering reviewers’ comments and recommendations.

Again I would like to express my sincere thanks and appreciations to all reviewers for their constructive suggestions and valuable directions which are being carefully addressed as shown in the following tables:

Reviewers’ comments and concerns – Reviewer (1):

Does the introduction provide sufficient background and include   all relevant references?

Improved

Is the research design appropriate?

Improved

Are the methods adequately described?

Improved

Are the results clearly presented?

Improved

Are the conclusions supported by the results?

Improved

Editor   Comment

Response

-   line 145 -  the sentence "Finally, in fifth phase, as..."   seems to be incomplete

Done   (line 139) , and

all   the similar mistakes are fixed along the paper.

-   line 189 -  "Query.Frequency ..." no dot is needed

Done   (line 186) , and

all   the similar mistakes are fixed along the paper.

-   line 191 -   "... query released ..." lacks the verb

Done   (line 188) , and

all   the similar mistakes are fixed along the paper.

-   line 258 -  "... as there is no site/cluster..." delete   "there is"

Done   (line 253) ), and all the similar mistakes are fixed along the paper.

-   line 290 -  the sentence is badly structured

Done   (line 285), and all the similar mistakes are fixed along the paper.

 Figures   10 and 11 -  the vertical axis is labelled TC in both figures, so what   do you mean by DDB Performance in Figure 11? I could not find a clear   definition of DDB Performance before; is it the average query response time??

Thank   you so much for your fruitful note which has been reflected in paper (lines   516-517):
  (DDBS performance which is calculated as (1- averaged TC), where averaged TC   is the averaged costs incurred as query set of certain experiment is being   processed. )

-   the bibliographical references are a mess. Some entries are listed with first   name first, some with first name initial, some with last name first.   Reference 13 has no title. In the text most references are listed by number,   but some by [Author, year]. In particular, I supposed that the most   referenced item (i. e. the previous work) is both referenced as [8] and [Adel   et al, 2017], but [Adel et al, 2017] could also correspond to [2] and   multiple references correspond to [Amer, 2018].

Done.   All references are fixed including title you referred to.

Any   appearance to [Adel et al, 2017] is replaced by [8].

In   conclusion I suggest the author either to produce a conference paper only   highlighting the improvements to the previous work or to re-write the paper   for a journal in a self consistent form (without continuously referring to   the previous paper)

Done

Paper   is re-written in a self-consistent form

Reviewer 2 Report

The work presents a clustering approach for distributed environments. The paper is well presented and the results were shown in a clearer way, with a comparison with related work.

My suggestions are:

- Please verify the spaces between paragraphs (some parts are different from others)

- Please verify the colors (yellow) in the tables. I suggest to change to other color (i.e. blue with bold font) 

Author Response

Dear respected Editor-in-chief and Unknown Reviewers,

I would like to thank you very much for giving me the opportunity to revise my manuscript based on your reviewers’ invaluable recommendations and suggestions. I am highly grateful for the time and energy they have put to review this work. I carefully went through their suggestions and made the necessary amendments to the manuscript considering reviewers’ comments and recommendations.

I would like to express my sincere thanks and appreciations to all reviewers for their constructive suggestions and valuable directions which are being carefully addressed as shown in the following tables:

Does the introduction provide sufficient background and include   all relevant references?

Improved

Are the conclusions supported by the results?

Improved

Drawback

Modification

-   Please verify the spaces between paragraphs (some parts are different from   others)

Done  

-   Please verify the colors (yellow) in the tables. I suggest to change to other   color (i.e. blue with bold font) 

Done

Reviewer 3 Report

This paper focuses on improving distribution performance based on transmission cost  minimization by using  data fragmentation and allocation techniques  beside several data replication scenarios and sites clustering algorithm.  
The presented experimental study shows the efficacy of the proposed approach on reducing Transmission Costs (TC). Three main contributions are presented in this paper : (1) improving the existing data fragmentation algorithm, (2) developing site clustering technique that aims to produce minimum number of high balanced cluster; (3) improving data allocation, and (4) studying the effectiveness of data replication impact on DDBS performance. 

The idea of the paper is promising. However, the structure of the paper should be checked. There are several subsections that could be merged (i.e. sections 7.6.x).   

The paper is an extended version of an already published paper (reference 8).  If so, the authors should clearly indicate the difference between this work and their previous paper (reference 8).

- The description of the clustering algorithm  in Section 5.3 could be formalized
- I suggest to add a table describing all the notations used in the sections 4, 5 and 6.
- The results presented in Section 7.3 should be interpreted and discussed.
- I really appreciate both the discussion part and the extended evaluation part.
- The writing style should be checked and a proofreading step by an English native speaker is advised.

Author Response

Dear respected Editor-in-chief and Unknown Reviewers,

I would like to thank you very much for giving me the opportunity to revise my manuscript based on your reviewers’ invaluable recommendations and suggestions. I am highly grateful for the time and energy they have put to review this work. I carefully went through their suggestions and made the necessary amendments to the manuscript considering reviewers’ comments and recommendations.

I would like to express my sincere thanks and appreciations to all reviewers for their constructive suggestions and valuable directions which are being carefully addressed as shown in the following tables:

Is the research design   appropriate?

Improved

Are the methods adequately described?

Improved

Are the results clearly presented?

Improved

Concern

Response

the   structure of the paper should be checked. There are several subsections that   could be merged (i.e. sections 7.6.x).  

Done.   Paper Structure is being modified.

The   paper is an extended version of an already published paper (reference   8).  If so, the authors should clearly indicate the difference between   this work and their previous paper (reference 8). 

Done   implicitly and explicitly in site clustering section, in the mixed-type   replication scenario and  in the results   and discussion sections

-   The description of the clustering algorithm  in Section 5.3 could be   formalized

Done.   Lines (238-242)

-   I suggest to add a table describing all the notations used in the sections 4,   5 and 6. 

Every   notation is defined in its first appearance which we see it would be better   so paper would not be that long and desecrating. I hope you would agree with   me on this point.

-   The results presented in Section 7.3 should be interpreted and   discussed. 

Done in lines (402-413   )

-   The writing style should be checked and a proofreading step by an English   native speaker is advised. 

Done
